# Phosphorylation of PACSIN2 at S313 Regulates Podocyte Architecture in Coordination with N-WASP

**DOI:** 10.3390/cells12111487

**Published:** 2023-05-27

**Authors:** Rim Bouslama, Vincent Dumont, Sonja Lindfors, Lassi Paavolainen, Jukka Tienari, Harry Nisen, Tuomas Mirtti, Moin A. Saleem, Daniel Gordin, Per-Henrik Groop, Shiro Suetsugu, Sanna Lehtonen

**Affiliations:** 1Research Program for Clinical and Molecular Metabolism, Faculty of Medicine, University of Helsinki, 00290 Helsinki, Finland; 2Institute for Molecular Medicine Finland (FIMM), Helsinki Institute of Life Science (HiLIFE), University of Helsinki, 00290 Helsinki, Finland; 3Department of Pathology, University of Helsinki, Helsinki, and Helsinki University Hospital, 05850 Hyvinkää, Finland; 4Department of Urology, Helsinki University Hospital, 00029 HUS, Finland; 5Department of Pathology, Helsinki University Hospital, 00290 Helsinki, Finland; 6Research Program in Systems Oncology, Faculty of Medicine, University of Helsinki, 00290 Helsinki, Finland; 7Children’s Renal Unit, Bristol Medical School, University of Bristol, Bristol BS8 1TS, UK; 8Minerva Foundation Institute for Medical Research, 00290 Helsinki, Finland; 9Abdominal Center, Nephrology, University of Helsinki and Helsinki University Hospital, 00290 Helsinki, Finland; 10Joslin Diabetes Center, Harvard Medical School, Boston, MA 02215, USA; 11Folkhälsan Institute of Genetics, Folkhälsan Research Center, 00290 Helsinki, Finland; 12Department of Nephrology, University of Helsinki, Helsinki, and Helsinki University Hospital, 00290 Helsinki, Finland; 13Department of Diabetes, Central Clinical School, Monash University, Melbourne, VIC 3800, Australia; 14Division of Biological Science, Graduate School of Science and Technology, Nara Institute of Science and Technology, Ikoma 630-0192, Japan; 15Data Science Center, Nara Institute of Science and Technology, Ikoma 630-0192, Japan; 16Center for Digital Green-Innovation, Nara Institute of Science and Technology, Ikoma 630-0192, Japan; 17Department of Pathology, University of Helsinki, 00290 Helsinki, Finland

**Keywords:** PACSIN2, syndapin2, actin cytoskeleton, podocyte, diabetic kidney disease, FFA, N-WASP

## Abstract

Changes in the dynamic architecture of podocytes, the glomerular epithelial cells, lead to kidney dysfunction. Previous studies on protein kinase C and casein kinase 2 substrates in neurons 2 (PACSIN2), a known regulator of endocytosis and cytoskeletal organization, reveal a connection between PACSIN2 and kidney pathogenesis. Here, we show that the phosphorylation of PACSIN2 at serine 313 (S313) is increased in the glomeruli of rats with diabetic kidney disease. We found that phosphorylation at S313 is associated with kidney dysfunction and increased free fatty acids rather than with high glucose and diabetes alone. Phosphorylation of PACSIN2 emerged as a dynamic process that fine-tunes cell morphology and cytoskeletal arrangement, in cooperation with the regulator of the actin cytoskeleton, Neural Wiskott–Aldrich syndrome protein (N-WASP). PACSIN2 phosphorylation decreased N-WASP degradation while N-WASP inhibition triggered PACSIN2 phosphorylation at S313. Functionally, pS313-PACSIN2 regulated actin cytoskeleton rearrangement depending on the type of cell injury and the signaling pathways involved. Collectively, this study indicates that N-WASP induces phosphorylation of PACSIN2 at S313, which serves as a mechanism whereby cells regulate active actin-related processes. The dynamic phosphorylation of S313 is needed to regulate cytoskeletal reorganization.

## 1. Introduction

Mammalian cells are highly complex self-organizing structures whose dynamic build is determined by numerous parameters such as size, shape, position and polarity, as well as internal and external signaling. The signaling mechanisms by which cells maintain this structure depend on the coordination between the cytoskeleton and proteins at the cell membrane. Podocytes, the glomerular visceral epithelial cells, perfectly illustrate the importance of the cytoskeleton in maintaining optimal cell function. In vivo, podocytes form interdigitating projections, called foot processes. The inter-cellular junctional structure joining adjacent foot processes is called the slit diaphragm. The slit diaphragm contains proteins typical of adherents and tight junctions [1,2] as well as unique podocyte proteins (e.g., nephrin) [3], which are connected to the actin cytoskeleton in order to maintain the functional integrity of podocytes [4]. During injury, for instance, diabetes and its complication of diabetic kidney disease (DKD), the rearrangement of the podocyte cytoskeleton leads to foot process flattening and retraction [5]. Since podocytes constitute an essential layer of the kidney filtration barrier, defects in their morphological stability and matrix adherence lead to kidney dysfunction.

Here, we sought to investigate the role of the actin-binding protein PACSIN2 (i.e., syndapin2) in regulating podocyte morphology and cytoskeletal arrangement and establish the biological relevance of its phosphorylation at S313 in podocytes. Recent reports link PACSIN2 to angiogenic sprouting [6], epithelial microvilli morphogenesis [7] and kidney tubule ciliogenesis [8]. These studies point out the role of PACSIN2 in the regulation of morphology and cell architecture. Furthermore, a study by Senju et al. brings to light the phosphorylation of PACSIN2 by protein kinase C α (PKCα) at serine 313 (S313), which can be triggered by cell detachment and shear stress [9]. Phosphorylation of PACSIN2 at S313 decreased its ability to bind to the plasma membrane and decreased the life span of caveolae [9]. However, the significance of S313 phosphorylation in various cellular processes, including cell spreading or survival, remains uncharacterized. We previously showed that the expression of PACSIN2 increases in glomeruli in DKD and accelerates the recycling of nephrin [10]. Despite this apparent association with glomerular pathophysiology, it remained unclear whether upregulation of PACSIN2 was beneficial in DKD and how it was triggered and regulated.

Relying on a rat model for obesity, diabetes and severe DKD, along with human nephrectomy samples and serum samples from individuals with type 2 diabetes (T2D), we show that the phosphorylation of PACSIN2 at S313 associates with increased circulating free fatty acids (FFA) and kidney dysfunction. Using cultured podocytes, we demonstrate that, in cooperation with N-WASP, PACSIN2 orchestrates architectural changes in podocytes, which can be influenced by the phosphorylation status of S313 and in turn impact the response of podocytes to injury.

## 2. Materials and Methods

### 2.1. Preparation of ZDF Rat and Human Glomerular Lysates

The isolation of glomeruli from ZDF rats (RRID: RGD_12859287) was performed using graded sieving with 250/150/75 µm sieves (Retsch, Haan, Germany) as previously described [10]. Human glomeruli were isolated from the non-malignant part of the kidneys from surgical nephrectomies performed at the Helsinki University Hospital using 425/250/150 µm sieves. Glomeruli were lysed in NP-40-based lysis buffer [11].

### 2.2. Western Blotting

Podocyte and glomerular lysates were analyzed by Western blotting as previously described [10], followed by imaging with the Odyssey^®^ CLx Imager (LI-COR, Lincoln, NE, USA) and quantification with Image Studio Lite 5.2 (LI-COR). The following antibodies were used: α-tubulin (T6199, mouse monoclonal), β-actin (A3853, mouse monoclonal, AB_262137), PACSIN2 (SAB1402538, mouse monoclonal) and Rac1 (05-389, mouse monoclonal) from Merck (Darmstadt, Germany), N-WASP (4848, rabbit monoclonal), RhoA (2117, rabbit monoclonal) and phospho-PKC Substrate (6967, rabbit mAb mix) from Cell Signaling Technology (Danvers, MA, USA), PACSIN2 (AP8088b, rabbit polyclonal) from ABGent (San Diego, CA, USA) and PKCα (AB11723, mouse monoclonal) from Abcam (Cambridge, UK). Antibodies for ubiquitin (sc177749, mouse monoclonal), FilaminA (sc177749, mouse monoclonal) and Dynamin2 (sc1666-69, mouse monoclonal) were purchased from Santa Cruz Biotechnology (San Diego, CA, USA). IRDye 800CW anti-rabbit IgG (926-32213, donkey) and IRDye 680RD anti-mouse IgG (926-68072, donkey) were from LI-COR (Lincoln, NE, USA). The antibody specific for PACSIN2 when phosphorylated at serine 313 (rabbit polyclonal) was described in reference [9]. Independent experiments were conducted on different days from cells with different passages. Each experiment was divided into separate replicates, treated and lysed separately. These replicates are displayed as individual data points.

### 2.3. Cell Culture and Preparation of Cell Lysates

Human podocytes, shown to be negative for mycoplasma, were maintained as described [12]. Shortly, proliferating podocytes were maintained in permissive conditions at 33 °C and thermo-switched to 37 °C to differentiate for 7–14 days. Sodium palmitate (Merck) was conjugated to FFA-free bovine serum albumin (BSA, Merck) at a 3:1 molar ratio at 37 °C for 1–2 h. When specified, fetal bovine serum in the medium was replaced by serum from individuals with T2D having either normal albumin excretion rate or moderate albuminuria, for 48 h before lysing the cells. The status of albuminuria was determined according to guidelines set by Nomenclature for kidney function and disease—executive summary and glossary from a Kidney Disease: Improving Global Outcomes (KDIGO) consensus conference [13]. High glucose treatment was performed by differentiating the cells for 10–14 days in a medium containing 30 mM glucose or 11 mM glucose and 19 mM mannitol as a control. Phorbol 12-myristate 13-acetate synthetic (PMA) treatment was performed on differentiated podocytes at 50 μM for 1 h. For palmitate treatment, the medium was supplemented for the specified length of time with 100 µM or 200 µM BSA-palmitate or BSA alone as a control, with or without bisindolylmaleimide (BIM, Merck) or dimethyl sulfoxide (DMSO) as a control. Cells were treated with 20 μM Wiskostatin (W2270, Merck) for 30 min, 50 μM tetracycline hydrochloride (T7660, Merck,) for 16 h, 10 μM LY294002 (S1105, Selleck Chemicals, Houston, TX, USA) for 16 h or 10 μM MMP-9 Inhibitor I (CAS 1177749-58-4—Calbiochem, Merck) for 16 h.

Proliferating podocytes were transiently transfected with flag-PKCα [9] or GFP-N-WASP [14] using Lipofectamine 2000 (Thermo Fisher Scientific, Waltham, MA, USA).

Cells were lysed in NP-40 or RIPA-buffer supplemented with protease and phosphatase inhibitors as described in [11,15].

### 2.4. FFA Measurements

Origin, maintenance, sacrifice and metabolic measurements of 8 and 34 weeks old lean and obese male ZDF-Leprfa/Crt rats have been previously described [10]. The FFA content of the serum of the ZDF rats and the individuals with T2D used to stimulate differentiated podocytes, were measured at the Biochemical Analysis Core for Experimental Research of the University of Helsinki using ADVIA 1650 (Siemens, Munich, Germany).

### 2.5. Apoptosis Assay

Proliferating podocytes were transfected with GFP-ev (empty vector), PACSIN2-wt, PACSIN2-S313E or PACSIN2-S313A using Lipofectamine 2000 [9]. After 72 h, cells were fixed with 4% PFA and stained with Annexin V-APC 1:50 (BD, Franklin Lakes, NJ, USA). The percentage of Annexin V-APC positive cells, in the GFP positive population, was measured by flow cytometry using BDaccuri (BD Life Sciences, Franklin Lakes, NJ, USA). A total of 10^5^ cells were detected in each sample.

### 2.6. Adhesion Assay

PACSIN2-wt or S313E/A cDNAs, described in [9], were subcloned into pCMV-myc vector (Cat. 631604, PT3282-5, Clontech, Mountain View, CA, USA) using KpnI and EcoRI sites. The constructs were verified by sequencing (see Appendix A). Proliferating podocytes were transfected with myc-ev, PACSIN2-wt, PACSIN2-S313E or PACSIN2-S313A using Lipofectamine 2000. After 48 h, cells were trypsinized and counted. Tissue culture 96-well plastic plates (Costar Corp., Cambridge, MA, USA) were coated with collagen IV or poly-L-Lysine for 1 h at 37 °C. Unspecific binding to the plates was blocked by incubating the wells with 1% BSA in PBS for 30 min at room temperature. 1 × 10^5^ cells were seeded to each well and incubated for 30 min at 37 °C. The wells were washed by means of immersion into a plastic tray containing PBS. Adhered cells were fixed with methanol and stained with Crystal Violet for 30 min (20% methanol, 0.1% Crystal Violet in H_2_O). After intense washings, cells were solubilized in 0.5% Triton X-100 and the number of cells was determined by measuring the absorbance at 595 mm using Hidex (Hidex, Turku, Finland).

### 2.7. Ubiquitination Assay

PACSIN2-wt or S313E/A were transiently transfected into proliferating human podocytes. Cells were lysed in NP-40 buffer supplemented with protease and phosphatase inhibitors. Lysates were precleared with protein A-Sepharose (Invitrogen, Waltham, MA, USA) and incubated at 4 °C for 16 h with anti-N-WASP antibodies or rabbit IgG (Invitrogen) as a control. The immune complexes were bound to protein A-Sepharose, washed with lysis buffer and immunoblotted as described above.

### 2.8. Immunofluorescence Analyses

PACSIN2-wt or S313E/A were transiently transfected into proliferating human podocytes. Cells were fixed with 4% PFA, permeabilized with 0.1% Triton-X100 and stained when mentioned with CellMask Blue (H32720, Thermo Fisher Scientific, Waltham, MA, USA), Hoechst (33342, Merck), anti-β-tubulin III IgG (T2200, rabbit polyclonal, AB_262133, Merck), phalloidin-488 (1:250, A12379, Thermo Fisher Scientific) to stain filamentous actin (F-actin), c-myc (M4439, mouse monoclonal, Merck,) and paxillin (610051, mouse monoclonal; BD Transduction laboratories, Franklin Lakes, NJ, USA) to stain focal adhesions. Alexa Fluor 594 anti-rabbit IgG (A-21207, donkey polyclonal, Invitrogen) and Alexa Fluor 488 anti-mouse IgG (A21202, donkey polyclonal, Thermo Fisher Scientific)) were used as secondary antibodies. Imaging was carried out using the Opera Phenix HCS system (PerkinElmer, Waltham, MA, USA) with a 20× air objective (NA 0.4), followed by processing with CellProfiler 3.1.8 (https://cellprofiler.org/ accessed on 10 January 2023) [16] to correct for non-uniform illumination, detect the cells or focal adhesions and extract numerical features.

### 2.9. Cell Classification

Advanced Cell Classifier (http://www.cellclassifier.org/ accessed on 12 December 2019) [17] was used to assign a phenotype to each cell by using the Multi-Layer Perceptron (MLP) supervised machine learning strategy where a non-linear model was trained from manually given annotations to classify each cell to a phenotype based on the numerical features measured with CellProfiler. To avoid technical bias, two independent models were trained on Advanced Cell Classifier to classify cells based on their (1) PACSIN2 overexpression and (2) overall cell morphology based on the actin cytoskeleton. For PACSIN2 overexpression, a model was trained to classify the cells into three phenotypes based on the intensity of myc staining: no, weak and high PACSIN2 overexpression. Only the numerical features for CellMask, Hoechst and c-myc were used to determine the over-expression phenotype. For cell morphology, another model was trained to classify the cells as “normal”, “altered” or rounded/dividing phenotype. The “normal” phenotype was defined as having well-organized and clear actin stress fibers whereas the “altered” phenotype included cells with disorganized actin or changed overall cell morphology based on the actin cytoskeleton. Only the numerical features for CellMask, Hoechst and phalloidin were used to determine the phenotype of the actin cytoskeleton and cell morphology. Based on the predictions using these two models, each cell received a class for the overexpression status and a class for the actin cytoskeleton and morphology status. The impact of myc-PACSIN2-wt/S313E/S313A overexpression was evaluated within individual coverslips by comparing the ratio of “normal” and “altered” cells amongst cells with high or no myc staining. Multiple features were analyzed for these two classes, using CellProfiler including those presented. FormFactor is calculated as 4 × π × Area/Perimeter2 and equals 1 for a perfectly circular object. Texture analyses measure intensity variation using the co-occurrence matrix.

### 2.10. Statistical Analyses

The statistical significance was calculated with GraphPad version 8.4.2 software (GraphPad Prism Software, La Jolla, CA, USA) and data were presented as mean ± SD. To compare differences between groups, we performed a Mann–Whitney test (two groups) or a one-way ANOVA test with Bonferroni post hoc test (multiple groups). For high-content analyzes, only a *p*-value of 0.001 or smaller was considered statistically significant.

## 3. Results

### 3.1. Phosphorylation of PACSIN2 at S313 Is Increased in the Glomeruli of Obese ZDF Rats

We observed that pS313-PACSIN2 was increased in the glomeruli isolated from obese ZDF rats at the age of 8 weeks compared to the glomeruli obtained from lean controls, but the ratio of pS313-PACSIN2 to total PACSIN2 was not increased (Figure 1A–D). In the glomeruli of 34-week-old obese rats, both the pS313-PACSIN2 and total level of PACSIN2 increased (Figure 1E–G), the latter also previously observed by our group [10]. Notably, the ratio of pS313-PACSIN2 to total PACSIN2 was increased in the glomeruli of 34-week-old rats (Figure 1H). These data suggest that diabetic conditions and potentially the progression of kidney disease, trigger phosphorylation of PACSIN2 at S313.

### 3.2. Phosphorylation of PACSIN2 at S313 Associates with DKD Rather Than with Diabetes

Next, we explored whether the phosphorylation of PACSIN2 at S313 was an early feature of diabetes, preceding the onset of albuminuria. We used human glomeruli isolated from individuals with T2D without DKD (their clinical characteristics are presented in Appendix A). We found that neither total nor pS313-PACSIN2 levels were changed in the glomeruli isolated from individuals with T2D, in comparison to individuals without diabetes (Figure 2A–D). Additionally, high glucose treatment of cultured differentiated human podocytes did not affect the phosphorylation of PACSIN2 at S313 or total PACSIN2 expression (Figure 2E–H). We hypothesized that diabetes and hyperglycemia alone were not sufficient to trigger the phosphorylation of PACSIN2. Therefore, we cultured differentiated human podocytes with serum from individuals with T2D, who have either normal albumin excretion rate or moderate albuminuria (their clinical characteristics are presented in Appendix A). The serum samples from the individuals with T2D and moderate albuminuria triggered a significant increase in pS313-PACSIN2 with a limited effect on total PACSIN2 or the ratio of pS313-PACSIN2 to total PACSIN2 when compared to podocytes treated with serum from individuals with T2D and normal albumin excretion rate (Figure 2I–L). These results suggest that the increase in total PACSIN2 and its phosphorylation at S313 are not a feature of diabetes per se but rather of the progression towards its complication, DKD. These data also propose that factors other than hyperglycemia lead to phosphorylation of PACSIN2 at S313 upon the development of DKD.

### 3.3. Phosphorylation of PACSIN2 at S313 Is Induced by Palmitate in a PKC-Dependent Manner

One of the factors contributing to kidney damage in DKD is dyslipidemia with concomitant accumulation of FFA in the bloodstream [18]. We, therefore, studied whether FFA triggers the phosphorylation of PACSIN2. We measured the FFA content in serum samples obtained from ZDF rats at 8 and 34 weeks of age and in serum samples of individuals with T2D (used in Figure 2I–L). As expected, the serum samples from the obese ZDF rats showed increased FFA levels compared to controls (Figure 3A,B). In the case of the serum from the individuals with T2D, the group with moderate albuminuria showed a trend of increase in FFA levels compared to the group with normal albumin excretion rate (Figure 3C). To confirm the link between FFA accumulation and phosphorylation of PACSIN2, we treated differentiated human podocytes with palmitate, the most abundant FFA in circulation. We found that palmitate significantly increased the phosphorylation of PACSIN2 at S313 (Figure 3D–G). We then hypothesized that the effects of palmitate on PACSIN2 phosphorylation were dependent on PKCα, which has been shown to phosphorylate PACSIN2 at S313 in Hela Cells (9). First, we addressed the question of whether PKCα mediates the phosphorylation of PACSIN2 at S313 in podocytes. We found that in the glomeruli of the 34-weeks-old obese ZDF rats when phosphorylation of PACSIN2 at S313 was the highest (Figure 1F), PKCα expression was increased (Figure 3H,I). Additionally, in differentiated human podocytes in culture, pharmacological activation of PKC with phorbol 12-myristate 13-acetate synthetic (PMA) drastically increased pS313-PACSIN2 and the ratio of pS313-PACSIN2 to total PACSIN2 (Figure 3J–M). Overexpression of PKCα in proliferating human podocytes increased both the expression of PACSIN2 and its phosphorylation at S313 without affecting the ratio of pS313-PACSIN2 to total PACSIN2 (Appendix A). These data indicate that PKC regulates the phosphorylation of PACSIN2 at S313 in podocytes. Based on these observations, we reasoned that inhibition of PKC could suppress the phosphorylation of PACSIN2 at S313 triggered upon palmitate treatment. We co-treated differentiated human podocytes with palmitate and the pharmacological inhibitor of PKC (bisindolylmaleimide, BIM). We found that, as anticipated, BIM prevented palmitate-induced phosphorylation of PACSIN2 at S313 (Figure 3N–Q). Taken together, our data imply that palmitate induces phosphorylation of PACSIN2 at S313 in podocytes through PKC.

### 3.4. PACSIN2 Overexpression Reduces Apoptosis and Enhances Adhesion Independently of the Phosphorylation Status at S313

Since pS313-PACSIN2 associates with kidney dysfunction and can be triggered by increased FFA, we examined whether phosphorylation of PACSIN2 was a protective or harmful mechanism in podocytes. In DKD, podocyte numbers are typically reduced due to detachment and/or apoptosis [5,19,20,21]. We overexpressed in proliferating podocytes GFP-PACSIN2-wild-type (GFP-PACSIN2-wt) and S313 phosphomimetic (GFP-PACSIN2-S313E) or non-phosphorylatable S313 (GFP-PACSIN2-S313A). We then measured the percentage of apoptotic annexin V-positive cells amongst the GFP-positive cell population (Figure 4A,B). We found that overexpression of GFP-PACSIN2-wt, GFP-PACSIN2-S313E and GFP-PACSIN2-S313A reduced apoptosis compared to the control GFP-empty-vector (GFP-ev). We also assessed podocyte adhesion to two different substrates (poly-L-lysine or collagen IV) after overexpression of myc-PACSIN2-wt, myc-PACSIN2-S313E and myc-PACSIN2-S313A. We found that overexpression of PACSIN2-wt increased adhesion to poly-L-lysine and collagen IV compared to the myc-ev (Figure 4C,D). The myc-PACSIN2 mutants showed a similar increase in reattachment compared to myc-ev, irrespective of the substrate. Intriguingly, we found no significant difference in adherence between the variants of PACSIN2. These experiments demonstrate that increased expression of PACSIN2 reduces podocyte apoptosis and improves adhesion regardless of the phosphorylation status of S313.

### 3.5. Phosphorylation of PACSIN2 at S313 Associates with Increased N-WASP Expression

To investigate the potential pathways affected by the phosphorylation of PACSIN2 at S313, we turned our attention to the interaction partners of PACSIN2 (Dynamin2 regulating trafficking and actin assembly and N-WASP, FilaminA, Rac1 and RhoA regulating actin cytoskeleton organization). Transient overexpression of PACSIN2-S313E in proliferating human podocytes induced an increase in N-WASP levels in comparison to empty vector (Figure 5A,B), without affecting the expression levels of Dynamin2, FilaminA, Rac1 and RhoA (Figure 5A,C–F). The increased N-WASP level following PACSIN2-S313E overexpression is likely due to decreased degradation of N-WASP, as we found that N-WASP immunoprecipitated from the lysates were less ubiquitinated when PACSIN2 phosphomimetic was overexpressed in cultured podocytes (Figure 5G,H). We also measured the expression level of N-WASP in the glomeruli isolated from obese ZDF rats at the ages of 8 and 34 weeks and found that N-WASP expression was increased compared to lean rats (Figure 5I–K). In glomerular lysates from individuals with T2D, we observed a trend of increase in N-WASP in comparison to controls (Appendix A). Together, these findings suggest that phosphorylation at S313 increases N-WASP expression, suggesting that S313 phosphorylation regulates actin-associated processes.

### 3.6. Phosphorylation of PACSIN2 at S313 Is Regulated by N-WASP Activity

Based on the observation that phosphorylation of PACSIN2 at S313 regulates N-WASP levels in podocytes, we reasoned that N-WASP could regulate the phosphorylation of PACSIN2 in return. Overexpression of N-WASP in cultured podocytes increased PACSIN2 expression (Appendix A) but did not affect pS313-PACSIN2 or the ratio of pS313-PACSIN2 to total PACSIN2 (Appendix A). We then treated differentiated human podocytes with the pharmacological inhibitor of N-WASP (wiskostatin). We found that wiskostatin increased the expression of PACSIN2 and drastically induced its phosphorylation at S313 (Figure 6A–D). This coincided with an increase in PKC activity (Figure 6E). We also observed that in cultured human podocytes, wiskostatin caused cells to retract (Figure 6F). We, therefore, reasoned that the phosphorylation status of PACSIN2 could be involved in these morphological changes. We overexpressed in proliferating human podocytes GFP-PACSIN2-wt, GFP-PACSIN2-S313E or GFP-PACSIN2-S313A and then measured cell size with and without wiskostatin treatment. We found that podocytes overexpressing GFP-PACSIN2-wt and -S313E remained significantly larger after wiskostatin treatment compared to cells overexpressing the control vector and treated with wiskostatin (Figure 6F–G, Appendix A). On the other hand, cells overexpressing non-phosphorylatable PACSIN2 sustained a notable decrease in size compared to cells overexpressing S313A treated with DMSO (Figure 6F–G, Appendix A). We also found that N-WASP inhibition altered GFP-PACSIN2 localization regardless of its phosphorylation status, causing it to form aggregates (Figure 6F). Taken together, these data suggest that N-WASP regulates PACSIN2 phosphorylation at S313 and that PACSIN2 and N-WASP cooperate to regulate the morphology of podocytes.

### 3.7. Dynamic Phosphorylation of PACSIN2 at S313 Is Required for Cell Spreading In Vitro

Analyses of wiskostatin-treated podocytes showed differences in cell size in podocytes overexpressing PACSIN2 without any treatment (Figure 6G). Additionally, in podocytes, N-WASP was previously shown to stabilize the actin cytoskeleton and maintain podocyte architecture [22]. We, therefore, reasoned that phosphorylation of PACSIN2 at S313 would affect actin-based processes such as cell morphology and cell spreading. We overexpressed in proliferating human podocytes GFP-PACSIN2-wt, GFP-PACSIN2-S313E or GFP-PACSIN2-S313A and then performed immunofluorescence-based high-content analysis followed by quantitative single-cell morphometry. We found that podocytes overexpressing GFP-PACSIN2-wt were larger than the cells overexpressing the control vector (Figure 7A,B, Appendix A) and possessed more focal adhesions (FA) per cell (Figure 7C,D). Overexpression of S313 phosphomimetic slightly increased cell size and both mutants of PACSIN2 slightly increased the FA number in comparison to GFP-ev, but less than overexpression of GFP-PACSIN2-wt (Figure 7A–D, Appendix A). These observations suggest that in podocytes, PACSIN2 regulates cell spreading.

We then tested whether the phosphorylation status of PACSIN2 at S313 would affect the response of podocytes to actin-related injury. We altered the actin cytoskeleton using cytochalasin D, which inhibits actin polymerization and assembly causing cells to retract. We then monitored the recovery after the washout (Figure 7E). We found that upon injury, cells retracted similarly regardless of PACSIN2 overexpression or its phosphorylation status (Figure 7F). However, during the expansion phase, podocytes overexpressing GFP-PACSIN2-wt showed a faster and more prominent increase in cell size compared to the control GFP-ev. Both mutants, S313E and S313A, behaved similarly to the control. Collectively, these data suggest that static phosphorylation or dephosphorylation of PACSIN2 at S313 halts its ability to regulate cell spreading. This is reinforced by the observation that during the washout of cytochalasin D in differentiated podocytes, endogenous levels of PACSIN2 initially increased at 15 min and then decreased at 90 min, while the phosphorylation at S313 was maintained at a constant level (Appendix A). Furthermore, we analyzed the effects of pharmacological inhibition of cell motility using three different compounds (tetracycline, LY294002 and MMP-9 inhibitor I, [23,24,25,26,27,28]) in proliferating podocytes. The ratio of pS313-PACSIN2 to total PACSIN2 increased after LY294002 treatment, decreased after MMP-9 inhibitor I treatment and remained stable after tetracycline treatment (Figure 7G–J). These data support the notion that both phosphorylation and dephosphorylation of PACSIN2 at S313 are involved in the regulation of cell spreading and motility, depending on the nature of the stimulus, the mechanisms at play and the kinetics of the pharmacological compound.

We speculated that the decreased cell spreading observed so far when overexpressing mutated forms of PACSIN2 in comparison to wt-PACSIN2 could be explained by differences in actin arrangement. We overexpressed in proliferating human podocytes myc-PACSIN2-wt, -S313E and -S313A; then we trained Advanced Cell Classifier to recognize their altered morphology (examples are shown in Figure 8A). We used cells not expressing PACSIN2 on the same coverslip as internal controls, to account for the potential effects of cell density and cell-cell interaction. We found that overexpression of myc-PACSIN2-S313E decreased the ratio of altered to normal cells in comparison to myc-PACSIN2-wt and myc-PACSIN2-S313A overexpression (Figure 8B). A closer look into the specific features distinguishing PACSIN2-S313E from -wt and -S313A shows that, compared to the internal control, overexpression of myc-PACSIN2-S313E limited the increase in cell rounding induced by PACSIN2 overexpression (Figure 8C) and reduced the increase in F-actin intensity and reorganization (Figure 8C–F). Taken together, these experiments indicate that, in podocytes, an increase in PACSIN2 initiates morphological alterations which can improve spreading by regulating actin arrangement and FA. Notably, for processes requiring active and rapid actin reorganization, dynamic phosphorylation is required.

## 4. Discussion

Our study shows that PACSIN2 is phosphorylated at S313 in DKD. We found that pS313-PACSIN2 increased with FFA accumulation, PKC activation and N-WASP inhibition. Our experiments highlight the reciprocity of the relationship between PACSIN2 and N-WASP as they coordinate to regulate cell architecture. Dynamic phosphorylation at S313 emerged as a regulatory mechanism during cytoskeletal reorganization. Since minor imbalances in podocyte cytoskeleton can disrupt the structural integrity of the glomerular filtration barrier, our data suggest that PACSIN2 could, via its phosphorylation at S313, be able to fine-tune the structure of the glomerular filtration barrier.

Previous studies have already linked PACSIN2 and N-WASP showing that PACSIN2 can directly interact with N-WASP and facilitate its recruitment to specific cellular locations, such as actin polymerization sites [30]. This ultimately influences actin dynamics by modulating N-WASP-mediated activation of Arp2/3, which initiates actin polymerization and branching [22]. Our study shows that phosphorylation of PACSIN2 at S313 regulates the interplay between PACSIN2 and N-WASP. Overexpression of S313 phosphomimetic in podocytes increased N-WASP levels (Figure 5A,B) and decreased N-WASP ubiquitination (Figure 5G,H). These data suggest that PACSIN2, through its phosphorylation at pS313, regulates N-WASP degradation. Suetsugu et al. have previously shown that N-WASP ubiquitination marks it for proteasomal degradation [14]. Since PACSIN2 directly interacts with N-WASP, it is possible that phosphorylation at S313 would stabilize the PACSIN2-N-WASP complex thereby preventing N-WASP recognition by its ubiquitin ligase or enhancing the association with deubiquitinating enzymes. In addition, it is plausible that phosphorylated PACSIN2 would alter N-WASP localization driving it away from proteasomes. Therefore, it seems likely that the increase in N-WASP observed in diabetic conditions (Figure 5I–K, Appendix A) could be at least partly induced by the phosphorylation of PACSIN2 at S313.

On the other hand, we found that N-WASP was able to regulate PACSIN2 phosphorylation as well. Pharmacological inhibition of N-WASP distinctively increased pS313-PACSIN2 and upregulated PKC activity (Figure 6A–E). Since phosphorylation of PACSIN2 increases N-WASP levels, this could be a compensatory mechanism in podocytes to offset the N-WASP blockade. A previous study has shown that PKC activation affects N-WASP localization, but is unclear how N-WASP activates PKC and in turn, increases pS313-PACSIN2 [31]. Further studies are needed to understand the mechanisms at play. Nevertheless, this suggests that N-WASP, PACSIN2 and PKC are tightly connected. In podocytes, we found that PACSIN2 phosphorylation at S313 was regulated by PKC (Figure 3J–Q), aligning with previous studies of pS313-PACSIN2 [9]. However, pharmacological inhibition of PKC alone, using BIM, did not affect the phosphorylation of PACSIN2 at S313 (Figure 3N–O). This suggests that the level of PKC inhibition achieved by BIM is not sufficient to completely block PKC activity. Alternatively, other kinases or signaling pathways (such as those regulated by N-WASP) could be compensating for the loss of PKC activity thereby restituting phosphorylation of PACSIN2 at S313.

In our investigations, it was clear that phosphorylation of PACSIN2 at S313 was not required for all PACSIN2 functions. For instance, N-WASP inhibition altered the localization of all three forms of GFP-PACSIN2 (Figure 6F) and phosphorylation at S313 did not affect podocyte survival or re-adhesion (Figure 4). The main difference we observed after overexpression of the phosphomimetic PACSIN2, non-phosphorylatable PACSIN2 or wt-PACSIN2 constructs was the regulation of cell size and actin re-arrangement. Our data highlight the complexity of the phosphorylation of PACSIN2 at S313 and its dependence on specific triggers. We found that overexpression of PACSIN2, regardless of its phosphorylation status, did not induce drastic changes in actin arrangement. However, high content analyses showed an increase in cell rounding and a change in F-actin structure upon overexpression of wt-PACSIN2 and non-phosphorylatable PACSIN2, in comparison to PACSIN2 S313 phosphomimetic (Figure 8C,D). These changes in structure might initiate preparedness for fast active actin-related processes, such as cell spreading and migration. Qualmann et al. observed in their early investigations of syndapin (aka PACSIN) that overexpression of syndapin induced morphological changes and increased cortical actin in HeLa cells, in accordance with our findings [32]. In the presence of a trigger requiring actin rearrangement, depending on the activated signaling pathways, PACSIN2 can be phosphorylated (Figure 6A–D, Figure 7G–J), dephosphorylated or not affected (Figure 7G–J). It is also very likely that the regulation of PACSIN2 and its phosphorylation in the presence of various triggers would be cyclical, as suggested by the cytochalasin D analysis in Appendix A. This would explain why static phosphorylation or dephosphorylation at S313 would impede cell spreading after cytochalasin D treatment (Figure 7F).

A few mechanisms could explain how PACSIN2 and its phosphorylation at S313 can regulate podocyte architecture. Primarily, our experiments emphasize the interconnection between N-WASP and PACSIN2. Although we show that non-phosphorylatable PACSIN2 increases cell shrinkage with wiskostatin treatment (Figure 6G), more studies are needed to establish how the phosphorylation status of PACSIN2 affects N-WASP localization and activation. Additionally, PACSIN2 overexpression in its different forms does not affect Dynamin2, FilaminA, Rac1 and RhoA at the protein level (Figure 5A,C–F). However, it is plausible that these proteins could instead have altered localization, activation or interactions with crucial partners. Moreover, PACSIN2 directly interacts with actin via its F-BAR domain [33]. PACSIN2 uses this same domain to bind membranes, depending on phosphorylation at 313 [9]. It is possible that the functions of the PACSIN2-actin complex vary depending on the phosphorylation status of PACSIN2 due to, for example, a conformational change. Furthermore, PACSINs interact with Rac1, Dynamin2 and N-WASP, all proteins implicated in cytoskeletal reorganization [34,35,36,37,38]. Although PACSINs use their Src-homology 3 (SH3) domain to recruit these partners, full-length PACSIN proteins are required for their proper function and localization [32]. Therefore, it is possible that phosphorylation at S313 affects the ability of PACSIN2 to recruit partners via the SH3 domain.

In the context of diabetes, our study presents new links in the signaling pathways regulating the development of DKD. Our experiments suggest that FFA accumulation triggers the phosphorylation of PACSIN2 at S313 via PKC and that pS313-PACSIN2 associates with the progression toward kidney disease. Given that PKCα activation is largely dependent on glucose concentrations in podocytes [39,40], it was surprising that PACSIN2 expression and phosphorylation at S313 did not associate with diabetes and hyperglycemia (Figure 2). This, however, matches the metabolic parameters of the ZDF rats at ages 8-weeks vs. 34-weeks (detailed in our previous study [10]) showing that phosphorylation of PACSIN2 at S313 precedes hyperglycemia. The notion that PACSIN2 phosphorylation at S313 follows increased FFA aligns with findings from our previous study, showing that PACSIN2 responds to palmitate [10], the most abundant circulating FFA [41]. This corroborates a previous report linking palmitate and PKC in podocytes, wherein palmitate increased endoplasmic reticulum stress which was reduced by PKC inhibition [42]. Furthermore, our findings suggest a new link between N-WASP and PKC (Figure 6A,E), which implies that the increase in N-WASP observed in DKD (Figure 5I–K, Appendix A) could affect PKC activity and ultimately pS313-PACSIN2.

Our study has limitations. Cell transfections were performed in proliferating podocytes as opposed to differentiated podocytes; as in our experience, modulations of PACSIN2 expression in differentiated cultured podocytes are quickly abolished. Therefore, some of the differences seen in our study could be connected to differences in podocyte proliferation, which could affect the present results.

## 5. Conclusions

Our study shows that PACSIN2 is phosphorylated at S313 in diabetes and DKD. By overexpressing PACSIN2 in cultured podocytes, we show that PACSIN2 could play a protective role in DKD, by decreasing apoptosis and reinforcing matrix adhesion. Phosphorylation of PACSIN2 at S313 emerged, in a complex interplay with N-WASP, as a dynamic process that could fine-tune the ability of PACSIN2 to modulate podocyte architecture. Our findings suggest that even mild cytoskeletal effects may have a substantial impact on the response of podocytes to injury, further emphasizing the complexity of the balancing forces behind cytoskeletal regulation. Future studies using knockout or transgenic PACSIN2 mice with experimentally induced DKD are needed to confirm whether changes in PACSIN2 levels affect podocyte survival and foot process effacement in vivo and understand the connection between PACSIN2, N-WASP and PKC.

## Figures and Tables

**Figure 1 cells-12-01487-f001:**
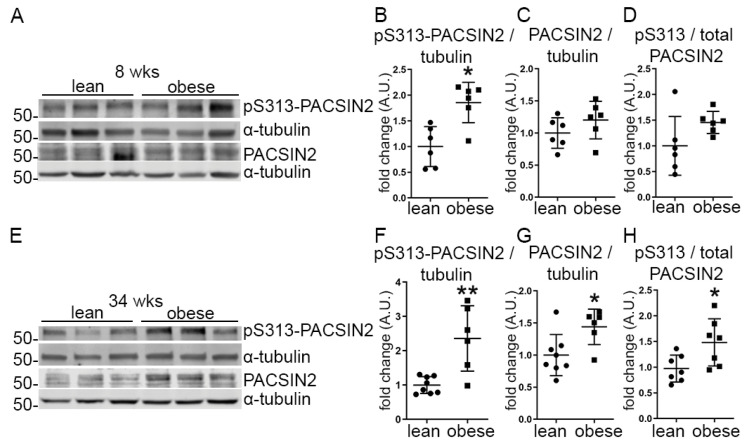
The phosphorylation of PACSIN2 at S313 is increased in the glomeruli isolated from obese ZDF rats. (**A**–**D**) Quantification of Western blots of total PACSIN2 and pS313-PACSIN2 (**A**) in lysates of glomeruli isolated from lean and obese ZDF rats at the age of 8 weeks shows increased phosphorylation of PACSIN2 normalized to α-tubulin (**B**) in the obese rats, but no difference in the ratio of pS313-PACSIN2 to total PACSIN2 (**D**). *n* = 6 for lean and obese. (**E**–**H**) Quantification of Western blots of total PACSIN2 and pS313-PACSIN2 (**E**) in lysates of glomeruli isolated from lean and obese ZDF rats at the age of 34 weeks shows increased expression and phosphorylation of PACSIN2 (**F**–**G**) in obese rats when normalized to α-tubulin. The ratio of pS313-PACSIN2 to total PACSIN2 is increased as well. (**H**) *n* = 8 for lean and *n* = 6 for obese. wks: weeks. *: *p* < 0.05, **: *p* < 0.01.

**Figure 2 cells-12-01487-f002:**
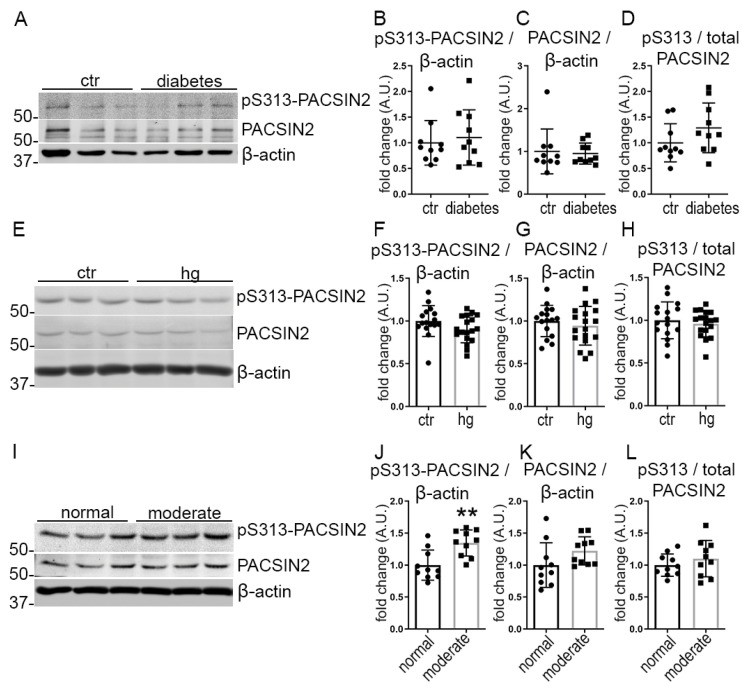
The phosphorylation of PACSIN2 at S313 associates with renal impairment in diabetes but not with diabetes alone. (**A**–**D**) Quantification of Western blots of total PACSIN2 and pS313-PACSIN2 (**A**) in lysates of glomeruli isolated from individuals with T2D shows no difference in the expression level of total PACSIN2 or phosphorylation at S313 normalized to β-actin in individuals with T2D compared to controls (**B**–**D**). *n* = 10 per group. (**E**–**H**) Quantification of Western blots of total PACSIN2 and pS313-PACSIN2 (**E**) in lysates of differentiated human podocytes treated with 30 mM glucose or mannitol as a control for 10–14 days of differentiation shows no difference in total PACSIN2 nor pS313-PACSIN2 normalized to β-actin (**F**–**H**). *n* = 18 from 3 independent experiments. (**I**–**L**) Quantification of Western blots of total PACSIN2 and pS313-PACSIN2 (**I**) in lysates of differentiated human podocytes treated with serum from individuals with T2D having normal albumin excretion rate or moderate albuminuria shows increased pS313-PACSIN2 normalized to β-actin (**J**) in podocytes treated with serum from individuals with T2D having moderate albuminuria. *n* = 10 per group. ctr: control, hg: high glucose. **: *p* < 0.01.

**Figure 3 cells-12-01487-f003:**
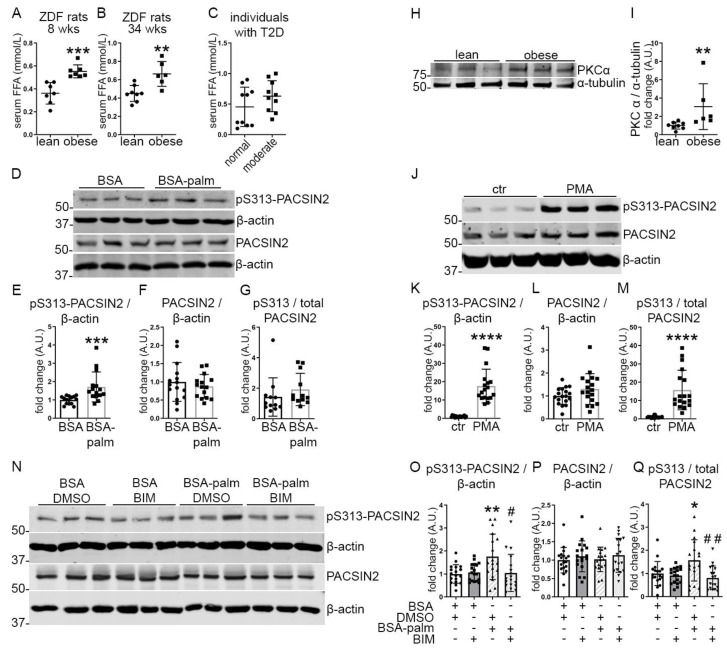
Palmitate stimulates the phosphorylation of PACSIN2 at S313 via PKC. (**A**–**B**) Quantification of FFA in the serum of 8-weeks-old and 34-weeks-old obese ZDF rats shows increased levels in comparison to lean controls of the same age. *n* = 7 for lean and obese in A, *n* = 8 for lean and *n* = 6 for obese in B. (**C**) Quantification of FFA in the serum from individuals with T2D and moderate albuminuria (the same serum samples were used in Figure 2I–L) shows a trend of increase in FFA levels in comparison to individuals with T2D and normal albuminuria. *n* = 10 per group. (**D**–**G**) Quantification of Western blots of total PACSIN2 and pS313-PACSIN2 (**D**) in lysates of differentiated human podocytes treated with 100 µM BSA-palmitate or BSA only as a control for 24 h shows increased pS313-PACSIN2 normalized to β-actin after incubation with BSA-palmitate (**E**). *n* = 15 from 3 independent experiments. (**H**,**I**) Quantification of Western blots of PKCα (**H**) in lysates of glomeruli isolated from lean and obese ZDF rats at the age of 34 weeks shows increased PKCα normalized to α-tubulin in the glomeruli of obese rats compared to lean controls of the same age (**I**). *n* = 8 for lean and *n* = 6 for obese. (**J**–**M**) Quantification of Western blots of total PACSIN2 and pS313-PACSIN2 (**J**) in lysates of differentiated human podocytes treated with 50 µM of PKC activator PMA for 1 h shows increased pS313-PACSIN2 normalized to β-actin, after PMA treatment compared to DMSO control (**K**). The ratio of pS313-PACSIN2 to total PACSIN2 is increased as well (**M**). *n* = 18 from 3 independent experiments. (**N**–**Q**) Quantification of Western blots of total PACSIN2 and pS313-PACSIN2 (**N**) in lysates of differentiated human podocytes co-treated with 200 µM BSA-palmitate and 200 ng/mL BIM, the pharmacological inhibitor of PKC, for 48 h shows that addition of BIM to cultures of podocytes treated with palmitate prevents the increase in pS313-PACSIN2 normalized to β-actin (**O**), as well as the ratio of pS313-PACSIN2 to total PACSIN2 (**Q**). BSA and DMSO were used as treatment controls. β-actin is used as a loading control. *n* = 18 from 3 independent experiments. * vs. BSA + DMSO. # vs. BSA-Palm + DMSO. wks: weeks, T2D: type 2 diabetes, palm: palmitate. *: *p* < 0.05, **: *p* < 0.01, ***: *p* < 0.001, ****: *p* < 0.0001, #: *p* < 0.05, ##: *p* < 0.01.

**Figure 4 cells-12-01487-f004:**
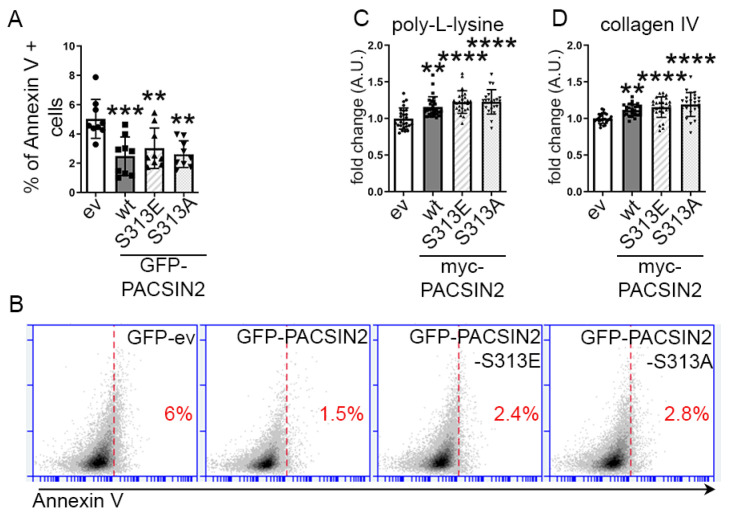
PACSIN2 overexpression is protective in podocytes regardless of its phosphorylation status at S313. (**A**,**B**) Flow cytometry of proliferating human podocytes after transient overexpression of GFP-ev, GFP-PACSIN2-wt, GFP-PACSIN2-S313E or GFP-PACSIN2-S313A and staining with annexin V shows that in the GFP-positive population, overexpression of all three forms of PACSIN2 decreases apoptosis in comparison to GFP-ev. *n* = 9 from 3 independent experiments. (**C**,**D**) Quantification of cells adhering to poly-L-lysine (**C**) or collagen IV (**D**) after trypsinization of proliferating human podocytes transiently overexpressing myc-ev or PACSIN2-wt/S313E/S313A shows that overexpression of all three forms of PACSIN2 increases reattachment in comparison to myc-ev, regardless of the substrate. Cell number was adjusted to 10^5^ before reseeding. *n* = 27 from 3 independent experiments. **: *p* < 0.01, ***: *p* < 0.001, ****: *p* < 0.0001.

**Figure 5 cells-12-01487-f005:**
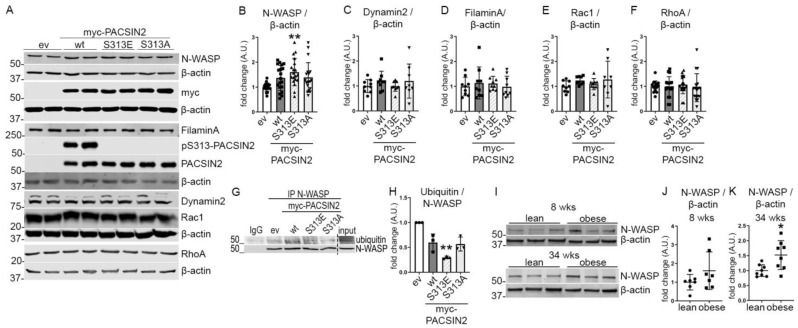
PACSIN2 phosphorylation regulates N-WASP expression. (**A**–**F**) Quantification of Western blots of total PACSIN2, pS313-PACSIN2, N-WASP, Dynamin2, FilaminA, Rac1 and RhoA (**A**) in lysates from proliferating human podocytes transiently overexpressing myc-ev or myc-PACSIN2-wt/S313E/S313A shows increased expression of N-WASP in podocytes overexpressing the three forms of PACSIN2 compared to ev control, with a significant increase in cells overexpressing S313E (**B**). PACSIN2 overexpression does not affect the expression of Dynamin2, FilaminA, Rac1 or RhoA (**C**–**F**). β-actin is used as a loading control. pS313-PACSIN2 blot shows that the antibody does not recognize PACSIN2 with mutations at S313. *n* = 15 from 5 independent experiments in **B**, *n* = 9 from 3 independent experiments in **C**–**E**, and *n* = 18 from 6 independent experiments in F. *: *p* < 0.05, **: *p* < 0.01 vs. ev. (**G**,**H**) Quantification of Western blots of ubiquitin in N-WASP immunoprecipitates from lysates of proliferating human podocytes transiently overexpressing myc-ev or myc-PACSIN2-wt/S313E/S313A shows decreased ubiquitination of N-WASP in cells overexpressing PACSIN2-S313E. *n* = 3 independent experiments. IP: immunoprecipitation. (**I**–**K**) Quantification of Western blot of N-WASP (**I**) in lysates of glomeruli isolated from lean and obese 8-week-old and 34-week-old ZDF rats shows increased N-WASP expression in comparison to lean controls (**J**–**K**). β-actin is used as a loading control. *n* = 7 in **J** and *n* = 8 in **K**. *: *p* < 0.05.

**Figure 6 cells-12-01487-f006:**
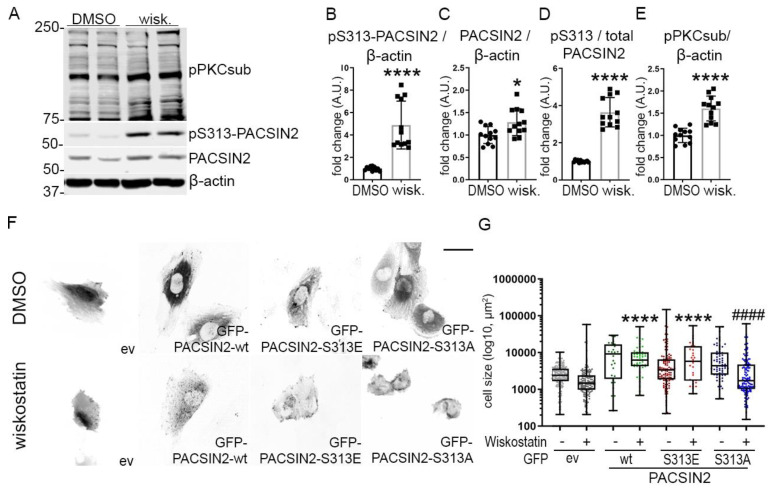
N-WASP regulates PACSIN2 phosphorylation. (**A**–**F**) Quantification of Western blots of total PACSIN2, pS313-PACSIN2 and pPKCsub, which recognizes phosphorylated substrates of PKC (**A**) in lysates of differentiated human podocytes treated with 20 µM N-WASP inhibitor wiskostatin for 30 min shows increased pS313-PACSIN2 (**B**), PACSIN2 (**C**) and the ratio of pS313-PACSIN2 to total PACSIN2 (**C**) compared to DMSO control. The activity of PKC is increased as well (**E**). β-actin is used as a loading control. *n* = 12 from 3 independent experiments. *: *p* < 0.05, ****: *p* < 0.0001. wisk.: wiskostatin. (**F**,**G**) Treatment of proliferating human podocytes transiently overexpressing GFP-ev or GFP-PACSIN2-wt/S313E/S313 with N-WASP inhibitor wiskostatin alters the localization of PACSIN2 regardless of its phosphorylation status (**F**). Podocytes overexpressing wt- and S313E-PACSIN2 do not shrink after wiskostatin treatment. ****: *p* < 0.0001 vs. ev wiskostatin. ####: *p* < 0.0001 vs. S313A-DMSO. Graph shows a representative experiment. *n* > 200. Experiment was repeated 4 times. scale bar = 20 µm.

**Figure 7 cells-12-01487-f007:**
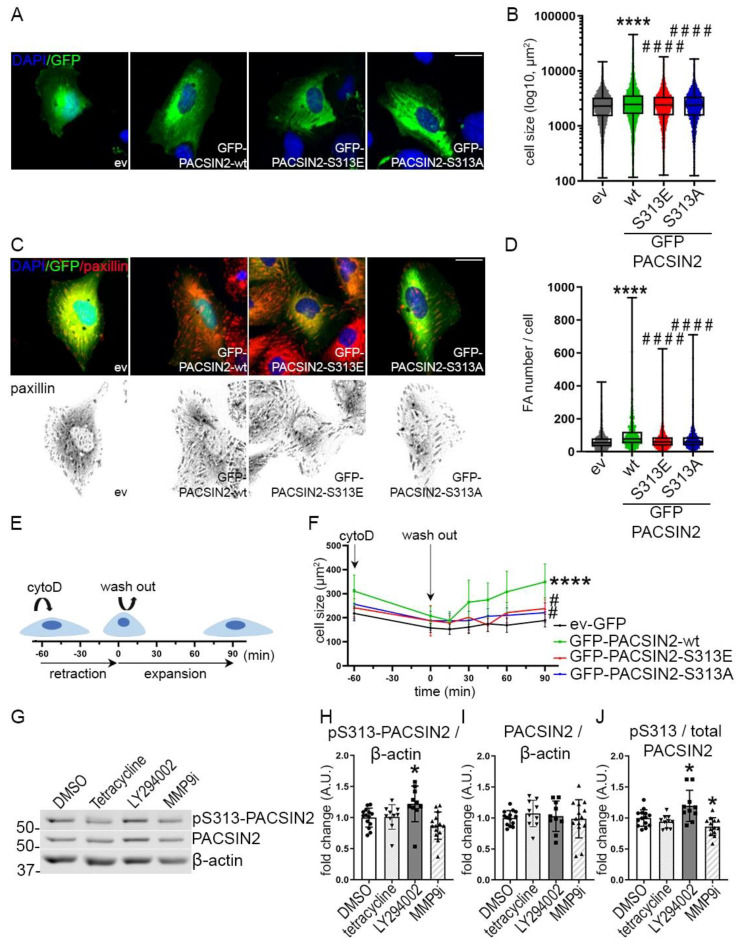
Phosphorylation of PACSIN2 at S313 affects podocyte spreading. (**A**,**B**) Quantification of the cell size of podocytes overexpressing GFP-ev or GFP-PACSIN2-wt/S313E/S313A (**A**) shows that cells overexpressing PACSIN2-wt display the most notable increase in size in comparison to GFP-ev or GFP-PACSIN2-S313E/S313A (**B**). (**C**,**D**) Analysis of focal adhesions from paxillin staining in podocytes overexpressing GFP-ev or GFP-PACSIN2-wt/S313E/S313A (**C**) shows increased number of focal adhesions in cells overexpressing PACSIN2-wt with minimal changes in cells overexpressing PACSIN2-S313E and -S313-A compared to ev (**D**). Graph shows a representative experiment. *n* > 200. Experiment was repeated 4 times. scale bar = 25 µm. DAPI is used for nuclear staining. (**E**,**F**) Treatment with cytochalasin D and washout of proliferating human podocytes transiently overexpressing GFP-ev or GFP-PACSIN2-wt/S313E/S313, as depicted in the cartoon in (**E**), shows that both the S313 phosphomimetic and the non-phosphorylatable S313 limit the improved spreading observed with GFP-PACSIN2-wt (**B**). *n* > 3150 from 4 independent experiments. Statistical significance was assessed by comparing the area under the curve for the slope of recovery using one-way ANOVA with a subsequent Bonferroni post hoc test. cytoD: cytochalasin D, FA: focal adhesion. ****: *p* < 0.0001 vs. ev-GFP. # *p* < 0.05 and ####: *p* < 0.0001 vs. wt. (**G**–**J**) Quantification of Western blots of total PACSIN2 and pS313-PACSIN2 (**G**) in lysates of proliferating human podocytes chemically treated to inhibit cell movement shows that the ratio of pS313-PACSIN2 to total PACSIN2 increases with LY294002, decreases with MMP-9 inhibitor I (MMP9i) and is not affected by tetracycline (**J**). *n* > 12 from 3 independent experiments. β-actin is used as a loading control. *: *p* < 0.05 vs. DMSO.

**Figure 8 cells-12-01487-f008:**
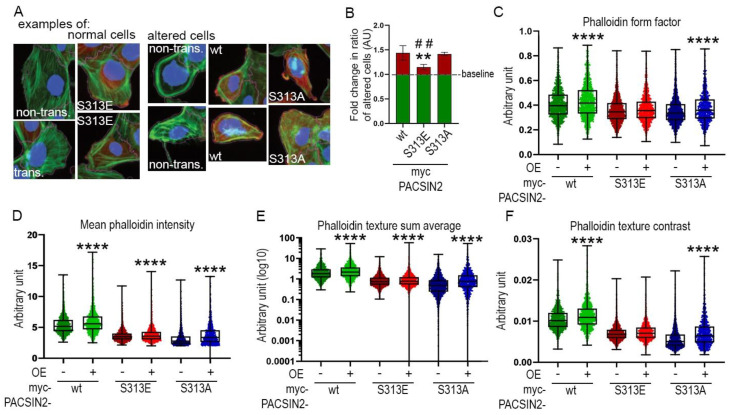
Morphometric analyses show that S313 phosphomimetic reduces cytoskeletal rearrangement in podocytes. (**A**) Examples of podocytes classified as having normal or altered morphology using Avanced Cell Classifier. Proliferating human podocytes transiently overexpressing myc-PACSIN2-wt/S313E/S313A were stained for F-actin in green (phalloidin), c-myc in red, whole cell (CellMask Blue) and nuclei (Hoechst) in blue, to assess cytoskeletal architecture. The fuchsia line marks the cell borders detected with CellProfiler. (**B**) Quantification of the ratio of altered to normal cells, examples of each shown in (**A**), shows that transient overexpression of myc-PACSIN2-S313E causes less cell alteration than PACSIN2-wt or -S313A. The baseline, set to 1, shows the basal ratio of altered to normal cells in non-overexpressing cells for each condition. *n* = 3 independent experiments. ** vs. wt. ## vs. S313A. (**C**–**F**) Features explaining the difference between the cells overexpressing PACSIN2 or the mutants and non-overexpressing cells classified by Advanced Cell Classifier. Box plots show that podocytes overexpressing PACSIN2-S313E display minimal changes in features such as “Phalloidin form factor”, “Mean phalloidin intensity” and “Phalloidin texture sum average” and “Phalloidin texture contrast”. Please note, that here “contrast” refers to one of the texture features defined by Haralick and colleagues [29]. ****: *p* < 0.0001.

## Data Availability

All data sets generated and analyzed during this study are included in this published article or its Appendix A. Additional data are available from the corresponding author upon reasonable request.

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
