# Peer review of "Phosphorylation of PACSIN2 at S313 Regulates Podocyte Architecture in Coordination with N-WASP"

_cells, 2023, doi:10.3390/cells12111487_

Round 1
Reviewer 1 Report
General comments
This report presents the results of an important study conducted with the protection of the glomerulus in mind, especially in difficult-to-treat renal diseases, and points to the possibility that PACSIN2-mediated regulation of the podocyte cytoskeleton may lead to protection of the glomerulus itself, or its function.
The data is abundant and much of it is detailed, indicating the magnitude of their effort, but I had some doubts noted below.
Each point
1. S313 Phosphorylation of PACSIN2 is presented under pathological conditions in this manuscript. Is it also observed under some physiological conditions? In other word, do the authors have any data to rule out the possibility that this phosphorylation is merely the “result” of various pathological reactions under pathological conditions and not a physiological regulatory mechanism important for cellular functions?
2. Is this phosphorylation catalyzed by PKC? Does "via" just mean that PKC might be involved? PMA exhibited the nice induction of the phosphorylation, while the induction by palmitic acid is remarkably weak. Additionally, substantial linkage between the phosphorylation and FFA is missing. In a correlation with 1., are there any possibility that the situation in which this phosphorylation occurs is only in an unusual state? If possible, for example, knockdown of PKC might help to address this question.
3. Moreover, the finding that increased adhesion and upregulation of N-WASP due to overexpression of PACSIN2 does not depend on this phosphorylation also raises doubts about the physiological significance of this phosphorylation.
4. The observation that constitutive S313-phosphorylated form of PACSIN2 only has weaker effect on cytoskeletal reorganization than that of the wild type PACSIN2 also makes me suspicious that this phosphorylation is in a consequence of pathology and may have poor physiological significance. If so, the “regulates” in the title of the manuscript might be too much and “participates” or “involves” might be suitable.
5. While I can agree that overexpression of wild-type PACSIN2 suppressing apoptosis is beneficial for cell survival, I am curious why the preservation of cell extensibility and the involvement of N-WASP can be concluded as beneficial.
Furthermore, if protection of cell extensibility is beneficial for cell survival, what does the inhibitory effect of phosphomutants to cell extensibility (it can be cytotoxic in this context) mean for the phosphorylation under pathological conditions? Does it mean that, for example, it is cytoprotective? If so, it can be much more important biological issue to be proved earlier rather than fine mechanisms of PACSIN2 upon the regulation of actin-cytoskeleton.
Author Response
We thank the Reviewers for the insightful review of our paper. Their constructive comments have helped to greatly improve the quality of the manuscript. Our detailed point-by-point responses to each of the comments are as follows:
Comment 1. The Reviewer is asking if the phosphorylation at S313 is only needed under pathological conditions.
It has been shown that phosphorylation of PACSIN2 at S313 increases with shear stress as well as with hypotonic stress (Senju et al. 2015). This study also emphasizes that phosphorylation of PACSIN2 at S313 regulates the life span of caveolae. All of these processes occur in healthy conditions, without necessarily progressing to pathology. In podocytes, it is our understanding that phosphorylation of PACSIN2 at S313 is a constant phenomenon, which can be upregulated and downregulated, as opposed to an ON and OFF system. This is why Western blots of lysates from healthy human kidney samples (Figure 2 A), healthy rat kidney samples (Figure 1 A,E), and control cultured podocytes (Figure 2 E,I) all show phosphorylation of PACSIN2 at S313. Additionally, to show that PACSIN2 phosphorylation is involved in cell contraction in non-pathological conditions, we inhibited cell movement and assessed pS313-PACSIN2 levels (new figure panels, Figure 7 GJ). We found that depending on the compound used, pS313-PACSIN2 was either increased, decreased or not affected. This finding supports the notion that both phosphorylation and dephosphorylation are involved in cell contraction. These new data are described in the Results section, lines 462-470.
Comment 2. The Reviewer is asking whether the phosphorylation of S313 is catalyzed by PKC.
Senju et al. showed that PKCα phosphorylates PACSIN2 at S313. In our study, we found that PKC is involved in the phosphorylation of PACSIN2 as both activation (Figure 3J-M) and inhibition (Figure 3N-Q) of PKC modulate pS313-PACSIN2 levels. Overexpression of PKCα in cultured proliferating podocytes increases PACSIN2 expression (Figure S1A, C) and phosphorylation (Figure S1A, B). These data suggest that PKCα regulates phosphorylation of PACSIN2 at S313. It is possible that multiple isoforms of PKC are involved in PACSIN2 phosphorylation, as well as other kinases. Notably, our novel data show that N-WASP inhibition as
well triggers the phosphorylation of PACSIN2 at S313 and the activation of PKC. The data is presented in Figure 6 A-D and is described in the Results section, lines 383-388.
Comment 3. The Reviewer points out that adhesion and N-WASP upregulation do not depend on phosphorylation doubting the biological relevance of this phosphorylation.
We have now repeated the experiment in Figure 5A, B which now shows clearer statistical significance for PACSIN2-S313E. The new Figure 6 shows that N-WASP inhibition leads to a drastic increase in pS313-PACSIN2 (Figure 6A-D) and that non-phosphorylatable S313A undergoes significant cell size reduction after N-WASP inhibition (Figure 6 G). Our study does not rule out that multiple effectors are involved in triggering PACSIN2 phosphorylation and regulating its function but emphasizes the complexity of this process. Our experiments suggest that pS313 could be a fine-tuning system rather than an ON-OFF switch.
Comment 4. The Reviewer is wondering whether the phosphorylation of S313 has limited effects on cytoskeletal organization in the absence of disease.
Our answer to this is partly discussed in comment 1. Our study ultimately emphasizes the complexity and the dynamism of the cytoskeletal regulation. Here, we also point out that although PACSIN2 has been shown to regulate the actin cytoskeleton and endocytosis, it is important to note that PACSIN2 knockout mice are healthy. This would suggest that the cells have compensatory mechanisms to circumvent the role of PACSIN2 or adjust when it is absent. This adds another layer of complexity in studying the phosphorylation of PACSIN2.
Comment 5. The Reviewer is asking why the enhanced cell spreading observed in WT-PACSIN2 is considered beneficial.
We believe that enhanced spreading can be beneficial in certain circumstances, for instance in response to changes in cell tension under certain physiological cues like shear stress or upon changes in the extracellular matrix stiffness. In the case of podocyte, widening of the foot processes of podocytes is a hallmark of glomerular injury. However, in the early stage of injury, this process is protective and aims to cover exposed glomerular basement membrane when neighboring foot processes of podocyte are lost due to detachment or apoptosis. However, we agree with the Reviewer that in absolute terms, increased cell spreading is not necessarily beneficial for cells. To reflect this more nuanced view, we have modified both the abstract and the discussion.
Reviewer 2 Report
Bouslama et al. present many data on the role of PACSIN2 in diabetes and podocyte biology, but not a coherent molecular mechanism or a clear disease relevance. More experiments are required to close at least some of the gaps.
1. Increased phosphorylation of PACSIN2 and increased total PACSIN2 levels are observed in obese ZDF rats but not in glomeruli of diabetic patients tested. Patient samples should be provided indicating that the alterations in PACSIN2 are indeed occurring in human T2D patients. Are the T2D patients tested in Fig. 2A-D only patients without moderate albuminuria?
2. What is “normal albuminuria”? T2D patients without albuminuria?
3. Apparently, about half of the individuals tested with “normal albuminuria” have FFA levels as high as the “moderate albuminuria” patients. If FFA is the crucial component in the serum responsible for PACSIN2 phosphorylation, one would expect a similar grouping in Fig. 2J-L. This is not the case.
4. PACSIN2 levels increased N-WASP protein levels. Since PACSIN and N-WASP were shown to bind to each other, this might be due to decreased N-WASP degradation on protein level.
5. Is PACSIN2 S313D showing a significantly larger increase of N-WASP than wt or S313A? Only in that case the authors can conclude that phosphorylation of PACSIN is of importance for the regulation of N-WASP levels.
6. For the experiments shown in Fig. 6B, D it will be important to show the averages of the independent experiments with statistics. Showing a single experiment can be quite misleading, since due to the high number of cells investigated even minute differences tend to be significant.
7. If N-WASP is playing a role in the observed cytoskeletal changes in response to PACSIN2 overexpression, why are N-WASP levels not correlating with the cytoskeletal changes? Similarly, changes observed in Fig.6 are not correlating to changes observed in Fig. 7. This rather suggests that the underlying mechanisms are not related.
8. The meaning of form factor and texture (Fig.7) needs to be explained.
9. How is “altered morphology” defined by Advanced Cell Classifier? What does it mean real morphological terms? How about showing whether cell area, cell form factor, and cell eccentricity are altered?
10. In Fig. 7, averages of independent experiments should be shown with statistical analysis.
11. If “OE” is overexpression, then “-“ would probably indicate the untransfected cells. Shouldn’t the values for the untransfected cells be the same for all experiments?
12. There is no functional experiment (siRNA, ko, N-WASP inhibitor) proving a linkage between the cytoskeletal effects and N-WASP. Would overexpression of N-WASP would have a similar effect?
13. Is N-WASP increased in obese rats or T2D patients?
Author Response
We thank the Reviewers for the insightful review of our paper. Their constructive comments have helped to greatly improve the quality of the manuscript. Our detailed point-by-point responses to each of the comments are as follows:
Comment 1. The Reviewer is asking about the phosphorylation of PACSIN2-S313 in the glomeruli of individuals with type 2 diabetes and whether samples in Fig. 2A-D are from patients with normal albumin excretion rate.
Samples in Figure 2A-D are from individuals with diabetes without diagnosed DKD. It is important to note that individuals with diagnosed diabetes are under medication to regulate their blood glucose levels. This is not the case for the obese diabetic ZDF rats, which makes these two set-ups widely different. Unfortunately, we do not have access to samples from individuals with DKD.
Comment 2. The Reviewer is asking about the meaning of “normal albuminuria”.
Here, we follow the most recent nomenclature for kidney function and disease (Levey et al., 2020) which encourages the use of albuminuria categories: Normal to mildly increased albumin to creatinine ratio is < 30 mg/g and moderately increased 30–300 mg/g. We thank the reviewer for pointing this out. This definition
has now been added to the Methods section (lines 120-123), including the reference.
Comment 3. The Reviewer points out that serum FFA levels in individuals with type 2 diabetes and moderate albumin excretion rate does not seem to correlate with pS313-PACSIN2 quantifications.
The grouping in FFA levels in Figure 3C does not match the grouping in Figure 2J, L suggesting that phosphorylation of PACSIN2 at S313 following FFA stimulation does not follow a simple linear dose-response pattern. It is also possible that other factors present in the serum influence the availability of FFA during the cell treatment and trigger competing mechanisms regulating the phosphorylation of PACSIN2.
Comment 4. The Reviewer suggests that PACSIN2 could be involved in regulating N-WASP degradation.
We thank the Reviewer for this suggestion. We have performed a N-WASP ubiquitination assay and found that overexpression of phosphomimetic S313E PACSIN2 decreases N-WASP ubiquitination (new Figure 5GH). The data is described in the Results section, lines 370-373.
Comment 5. The Reviewer is asking whether overexpression of PACSIN2-S313E induces a significantly larger increase of N-WASP in comparison to PACSIN2-WT or PACSIN2-S313A.
As noted in response to comment 3 from Reviewer #1, we have added data points to the quantification of NWASP expression after overexpression of PACSIN2 which has now changed the statistical significance of the findings (Figure 5B).
Comment 6. The Reviewer would like the data in Figure 7 B to be displayed as averages of independent experiments with statistics because a high number of data points may give significant results even with minute changes.
We agree with the Reviewer that increased number of data points increases statistical power and can highlight subtle changes. However, one of the advantages of high-content analyses is to detect subtle changes in cells, as these are usually precursors to the drastic and sometimes catastrophic changes. Moreover, using averages of independent experiments dismisses the importance of the distribution of the data which also conveys important information. Additionally, in our experience, a high number of data points does not necessarily imply significance. Nonetheless, to address the Reviewer’s concern, as in Calizo et al. 2019, only a p-value <0.001 is now considered statistically significant for high content analyses (see page 5, lines 217-218). We would also like to point out that our analyses found no significant differences between the different forms of PACSIN2 in several morphological measurements.
Comment 7. The Reviewer suggests that the mechanisms behind the changes observed in Figure 7 and Figure 8 could be unrelated, and do not correlate with N-WASP expression.
We agree with the Reviewer that the phosphorylation of PACSIN2 at S313 seems complex and its regulation seems multifactorial, which adds to the intricacy of this system. We found that overexpression of PACSIN2 phosphomimetic increases N-WASP level (Figure 5 A, B) by decreasing its degradation (Figure 5 G, H). Based
on this correlation, we would expect that inhibition of N-WASP would lead to dephosphorylation of PACSIN2. However, N-WASP inhibitor wiskostatin treatment clearly triggered the phosphorylation of PACSIN2 at S313 (Figure 6 A-D). Our study aims to emphasize the link between N-WASP function and PASIN2 function. We
believe that this reciprocal regulation is highly dependent on the type of trigger and its kinetics, making it difficult to draw extrapolation from different set-ups.
Comment 8. The Reviewer asks for the meaning of form factor and texture in Figure 8.
We thank the Reviewer for bringing this to our attention. These have been added to the Methods section (line 209-211).
Comment 9. The Reviewer asks how the classification of cells in Figure 8 is carried out and suggests using various measurable features.
Cell classification is detailed in the methods section (lines 189- 208). Briefly, Advanced Cell Classifier was trained to recognize PACSIN2 overexpression levels based on c-myc staining and altered cells based on overall morphology and disorganized actin. Features with significant differences between the different forms of PACSIN2 were displayed in Figure 7 and Figure 8. These include features suggested by the Reviewer (cell area in Figure 7B and cell form factor in Figure 8B). We have now noted in the Methods section that other feature analyses have been carried out as well (lines 208-209).
Comment 10. The Reviewer asks for averages of independent experiments to be used for experiments from Figure 8.
This concern was addressed in the response to comment 6.
Comment 11. The Reviewer points out that in Figure 8 the untransfected cells have different values in the different conditions.
We believe that the variation in the values for the control untransfected cells vary most likely due to differences in cell confluence between the different conditions, which can influence morphological features.
Comment 12. The Reviewer asks for a functional experiment linking N-WASP to the cytoskeletal effects measured in this study.
We thank the Reviewer for this suggestion. We have now added a supplementary figure showing that NWASP overexpression increases PACSIN2 with no effect on pS313-PACSIN2 and the ration of pS313-PACSIN2 to total PACSIN2 (Figure S3). In the newly added Figure 6, we show that inhibition of N-WASP triggers the phosphorylation of PACSIN2 at S313. As we noted that podocytes retract in response to N-WASP inhibition, we have used this treatment concomitantly with PACSIN2 overexpression in cultured podocytes. We found that cells overexpressing non-phosphorylatable PACSIN2 undergo a significant decrease in size (Figure 6G)
The data is described the Results section, lines 380-402.
Comment 13. The Reviewer asks whether N-WASP is increased in individuals with type 2 diabetes and in obese ZDF rats.
Figure 5 I-K shows that N-WASP is increased in the glomeruli of obese diabetic ZDF rats at the ages of 8 and 34 weeks. In the glomeruli of individuals with type 2 diabetes, there was a trend of increase in N-WASP now shown in Figure S2. The data is described in the Results section, lines 373-377.
Reviewer 3 Report
Following on the lab's previous findings regarding the role of PACSIN2 in kidney disease (FASEB Jrnl. 2017), the current study by Bouslama et al examines the potential significance of phosphorylation of PACSIN2 in relevant kidney model animals and cultured cells. This study demonstrates that increased phosphorylation of PACSIN2 occurs at Serine 313 (pS313) in the glomeruli of diabetic rats and in cultured human podocytes exposed to serum from diabetic patients with albuminuria. They also provide compelling evidence regarding the factors regulating this phosphorylation. The study then examines the functional significance of pS313 on various aspects of podocyte behavior. The experiments are well-designed and controlled, thoroughly quantified and analyzed, and appropriately interpreted. Given the importance of podocytes to many forms of kidney disease, I believe this study will be of interest to both clinicians and scientists in the kidney field. This work is, in my opinion, appropriate for publication in Cells, though I have several questions and comments that I hope the authors can first address.
1. The text describing the increased total PACSIN2 and pPACSIN2/total PACSIN2 levels includes reference to graphs 1C and 1D, but those are the 8wk animals, and the comparisons did not reach statistical significance. It seems appropriate then to note that the increases in those measures were only significantly different at 34wks.
2. In section 3.2 of Results, the authors report no increase in pS313 in glomeruli of patients with T2D but without DKD. This is appropriate for their question of whether phosphorylation precedes albuminuria. However, based on their developing model that PACSIN2 phosphorylation is part of a protective response as kidney disease progresses, it would be interesting to examine pS313 levels in glomerular lysates from T2D patients with DKD. This does not need to be done for acceptance of this manuscript, in my opinion, but if it is easily accomplished, it may strengthen their model.
3. Why are the apoptosis experiments conducted on proliferating podocytes, without differentiation? This would seem to open the possibility that PACSIN2 manipulations could be affecting mitosis, rather than some unique aspect of podocytes, which are of course fully differentiated, post-mitotic cells. It is my understanding that without the shift to 37C for differentiation, these cells do not express the podocyte-specific slit diaphragm and cytoskeletal adapter proteins that make them a relevant model.
On a related note, the experiments examining the effects on cell size, shape and actin organization (Figures 6 and 7) are also conducted in undifferentiated, proliferating cells. Is there a reason for this? Since these cells are actively dividing, the cell cycle will undoubtedly cause changes in cell size, shape and actin organization. Thus, the differences noted in the different PACSIN2 variants could be due to changes in proliferation rates/cell cycle progression, rather than more directly on the cytoskeleton; the two are not mutually exclusive of course. It would seem that performing this type of experiment in differentiated podocytes would be more relevant. While I do not feel that all these experiments need to be repeated in differentiated cells, if there is a reason why they were done in proliferating cells, it would be good to include that explanation, and it would also seem appropriate to mention the possibility that there could be differences in proliferation in these cells that might affect the phenotypes being analyzed.
4. Can the authors speculate as to why BIM treatment alone had no apparent effect on basal pS313 levels (Figure 3 N,O).
5. Can the authors please provide more information (in the text, figure legends, or methods) describing exactly what each data point in the various graphs represents in terms of experimental replicates. Most notably, in some of the quantifications of Western blots, it is not obvious to me what each data point represents. For example, in Figure E-H, how exactly are 18 data points being derived from 3 independent experiments? Were 6 Westerns run with the same 3 lysates from the 3 experimental replicates, or some other arrangement? More information on precisely what each data point represents experimentally would be helpful.
6. In some of the Western blot images, the loading control signal appears saturated (for example, Figure 3J and N), which would limit the ability to detect differences in those protein levels and skew the normalizations for the proteins of interest. I believe the dynamic range of the LICOR Odyssey is quite large, so perhaps this is not an issue, but have the authors confirmed that those signals are not saturated?
7. In section 3.5, the text notes, "Transient overexpression of PACSIN2-wt, -S313E or -S313A S313A in proliferating human podocytes all induced an increase in N-WASP levels (Figure 5A-B)…", however there is no indicator of statistical significance in the graph for S313A. If this comparison to ev is not significant, it seems the text needs to be modified to indicate this. If it is not different, I believe this would actually seem to support the author's conclusion that phosphorylation at S313 is important in driving N-WASP upregulation. (Also, note the duplication of "S313A" in the cited text, which should be corrected.)
8. Perhaps it is just the proofs, but the resolution is very low in some of the figures. Hopefully this will be improved in a final draft.
9. There are no functional data regarding the significance of N-WASP upregulation following PACSIN2 overexpression, and the interpretation of this upregulation is appropriately kept fairly speculative in the main text. However, the language in the abstract may be a bit overstated for the current data, "...overexpression of PACSIN2 is beneficial in podocytes apparently via regulation of N-WASP…". Although this is a reasonable hypothesis, in the absence of functional evidence, I would recommend softening the conclusion here to indicate that the current data are simply consistent with a hypothesis in which upregulated N-WASP may be an important mediator of increased PACSIN2 levels.
Somewhat related to this, the conclusion that other potential interactors of PACSIN2 are not affected (i.e., Dynamin, FilaminA, Rac1, RhoA) is based on their total protein levels, however, it is possible that their activity or localization is affected, independent of total protein levels. The authors do discuss a possible effect on FilaminA localization in the Discussion, but perhaps these types of potential effects of PACSIN2 on other known interactors could also be mentioned.
10. I don't believe the Methods section mentions the type and source of secondary antibodies used in the Western blotting and immunofluorescence experiments. These should be included.
11. In section 3.4, the PACSIN2 mutants S313E and S313A are described as "constitutively phosphorylated or dephosphorylated PACSIN2", however, I believe it is more accurate to describe these mutants as phosphomimetic and non-phosphorylatable.
12. In Figure 6F, the quantification of cell size following cytoD treatment and washout is statistically compared using the area under the curve, however this measure would seem to be confounded by the size of the cells in general, which are larger in the PACSIN2 wt cells. Since the experiment seeks to examine the effect of these proteins on the regulation of actin polymerization, and subsequently cell size, following the washout, it would seem that the better comparison might be the slope of the lines following washout. I suspect the wt-PACSIN2 will still be significantly different from the others, but if they authors agree, it may be a more appropriate test.
13. Admittedly I have no experience plotting the type of high content data presented in Figure 6B and D, but is it standard in the field to have the box and whisker plots cover the entire range of data points and overlay the actual data points? Many of the data points end up obscured in this presentation. Can mean and standard deviation plots, or some similar box plots be used that still represent the distributions effectively, while enabling visualization of the data points themselves?
14. There should also be some basic descriptive statistics provided (such as mean and standard deviation) for some of the experiments. For example, cell size and focal adhesion number (Figure 6B,D).
15. Line 412 refers to Figure 6I, but there is no panel I in Figure 6.
16. Figure 7, the images in panel A or the figure legend should indicate which color channel is labeling which stain. Also, the legend mentions a fuchsia line demarcating the cell borders, but this not readily visible in my proof, perhaps because the resolution is quite low. Hopefully this will be more apparent in final version.
Author Response
We thank the Reviewers for the insightful review of our paper. Their constructive comments have helped to greatly improve the quality of the manuscript. Our detailed point-by-point responses to each of the comments are as follows:
Comment 1. The Reviewer wishes us to only describe statistically significant changes in Figure 1.
This was corrected in the Results section, page 5, lines 221-228.
Comment 2. The Reviewer asks us to examine phosphorylation of PACSIN2 at S313 in in glomerular lysates from T2D individuals with DKD.
We agree with the Reviewer that this information would strengthen the study. However, none of the patients from which our nephrectomy samples were taken has been diagnosed with DKD.
Comment 3. The Reviewer asks why we have conducted some of the experiments on proliferating podocytes.
Unfortunately, the choice of proliferating podocyte is due to the difficulty in modulating PACSIN2 in cultured podocytes. We have tried establishing both a stable cell line of podocytes overexpressing PACSIN2 ortransiently overexpressing PACSIN2 in differentiated podocytes, using different systems. Differentiated
podocytes in culture return to a “normal” expression level of PACSIN2 very quickly. It is unclear to us why this is so. We have now added this limitation to the Discussion section (line 621-625).
Comment 4. The Reviewer asks us to speculate on why BIM treatment alone had no apparent effect on basal pS313 levels (Figure 3 N,O).
In Figure 3 N,O, BIM alone did not decrease pS313-PACSIN2 levels. This could indicate that the level of PKC inhibition achieved by BIM, in the absence of a trigger, is not sufficient to completely block PKC activity. Alternatively, there might be other kinases or signaling pathways that can compensate for the loss of PKC
activity and phosphorylate PACSIN2 at S313. This has now been clarified into the Discussion section (lines 559-564).
Comment 5. The Reviewer asks us to describe exactly what each data point refers to in each graph.
For Western blots, each data point represents a separately treated condition and combines conditions from independent experiments. This means that if the graph shows 9 data points from 3 independent experiments,
each experiment contained 3 separately treated conditions. For high content data, each data point represents a cell and the graph shows one representative experiment. This has now been clarified in the Methods section (lines 109-112).
Comment 6. The Reviewer asks us to confirm that signals of some of the loading controls are not saturated.
The majority of the blots were scanned with Odyssey® CLx Imager using its automatic mode which “acquires images with virtually no saturated pixels on the first attempt with no user adjustments”. We reference here the Imager’s manual (https://licor.app.boxenterprise.net/s/8prh6ps2abjbemx68412, p18). Additionally, Image Studio Lite 5.2, the software used to quantify signal Western blot does not allow quantification of area with saturated pixels.
Comment 7. The Reviewer points out the discrepancy between the text and the statistics displayed in Figure 5 B.
We thank the Reviewer for their diligence. Our previous submission mistakenly omitted the indicator of statistical significance in Figure 5B for S313A. However, the statistical analysis has changed with the replication of the experiment. This was explained in response to comment 3 from Reviewer #1.
Comment 8. The Reviewer points out the low resolution in some of the figures.
We are not sure why the resolution of the figures would be low for the Reviewer. This could be technical matter related to the submission process.
Comment 9. The Reviewer remarks the lack of functional data regarding N-WASP upregulation following PACSIN2 overexpression and asks to note potential effects of PACSIN2 on other known interactors beyond protein expression.
To strengthen the link between N-WASP and PACSIN2 phosphorylation, a few experiments have been added:
- PACSIN2 phosphorylation decreases N-WASP degradation (Figure 5G, H).
- Inhibition of N-WASP triggers the phosphorylation of PACSIN2 (Figure 6A, D).
- Increased PACSIN2 and phosphomimetic inhibit cell retraction when cells are treated with N-WASP inhibitor (Figure 6F, G).
These new data are described in the Results section, lines 367-402. We have also noted in the Discussion section (lines 593-595) that PACSIN2 overexpression can affect the localization or activation of its partners as well.
Comment 10. The Reviewer asks us to mention the type and source of secondary antibodies used in the Western blotting and immunofluorescence experiments.
This has now been added to the Methods section (lines 106-108, 180-182).
Comment 11. The Reviewer believes that it is more accurate to use the terms “phosphomimetic” and “nonphosphorylatable”.
We agree. This has been changed throughout the manuscript.
Comment 12. The Reviewer suggests to limit the statistical analysis in Figure 6F to the slope of recovery.
We agree. This has now been noted in the figure legend (line 484).
Comment 13. The Reviewer asks about the choice of graph for the high content experiments.
The choice of the graph is in line with previous studies of this type (Calizo et al., 2019) and aims to illustrate different measures in the data set. However, to address the issue of visibility, we have changed the colors of graphs and their dot plots and increased the size of the graphs.
Comment 14. The Reviewer asks for descriptive statistics for some of the experiments.
Descriptive statistics of some of the high content analyses have now been added to Supplementary material (Tables S3-S5).
Comment 15. The Reviewer notes reference to a non-existing Figure (Line 499).
This has now been corrected (line 499).
Round 2
Reviewer 1 Report
I got sincere reply from authors.
They properly responded to the questions I made.
Reviewer 2 Report
The authors addressed all issues raised sufficiently.